# Design of a Multiple Folded-Beam Disk Resonator with High Quality Factor

**DOI:** 10.3390/mi13091468

**Published:** 2022-09-04

**Authors:** Xiaopeng Sun, Xin Zhou, Lei Yu, Kaixuan He, Dingbang Xiao, Xuezhong Wu

**Affiliations:** 1College of Intelligence Science and Technology, National University of Defense Technology, Changsha 410073, China; 2East China Institute of Photo-Electronic IC, Bengbu 233042, China

**Keywords:** microelectromechanical systems, quality factor, thermoelastic dissipation, anchor loss, disk resonator

## Abstract

This paper proposes a new multiple folded-beam disk resonator whose thermoelastic quality factor is significantly improved by appropriately reducing the beam width and introducing integral-designed lumped masses. The quality factor of the fabricated resonator with (100) single crystal silicon reaches 710 k, proving to be a record in silicon disk resonators. Meanwhile, a small initial frequency split of the order-3 working modes endows the resonator with great potential for microelectromechanical systems (MEMS) gyroscopes application. Moreover, the experimental quality factor of resonators with different beam widths and relevant temperature experiment indicate that the dominating damping mechanism of the multiple folded-beam disk resonator is no longer thermoelastic damping.

## 1. Introduction

Microelectromechanical disk resonators exhibit tremendous potentials in many transduction applications [1,2,3,4], benefitting from their symmetrical structure and mature fabrication process. The mechanical quality factor (*Q*) of the resonator, which describes the ratio of the totally stored energy to the dissipated energy per vibration cycle, is directly related to the sensitivity and signal-to-noise ratio of the transducer devices. There are different sources of energy dissipation in MEMS resonators, including viscous air damping, thermoelastic damping (TED), anchor loss, surface loss, phonon-phonon scattering, and intrinsic material loss [5,6]. Different kinds of dissipation mechanisms are treated as paralleling dampers, and the overall *Q* is determined by ∑*Q*^−1^ = ∑*Qi*^−1^, where *i* labels different mechanisms. When air damping is minimized through high vacuum packaging, energy dissipation in most disk resonators is mainly governed by the TED [7,8].

TED is related to the coupling of thermal and elastic deformation fields of the resonator through the coefficient of thermal expansion of the material. Using high-quality non-silicon material is one promising approach but it faces manufacturing challenges [9,10]. Thus, optimizing the structure design of silicon MEMS resonators to improve the quality factor is still attractive due to the mature processing technology. The reported *Q*-enhancing methods for disk or ring resonators include optimizing the thickness distribution [1], introducing slots [11], adding lumped masses to the flexural body [1,12], and many other topological optimization methods [13,14,15,16]. Among them, adding lumped masses to realize stiffness-mass decoupling has an obvious improvement effect. However, the existing arrangement of masses in disk resonators is limited to hanging or attaching to the flexible rings, which is not conducive to improving the effective capacitance area and the upper limit of performance [14]. The radially pleated disk resonator reported in [15] shows great potential in the quality factor even without introducing lumped masses design, which has advantages in vibration amplitudes over other topological forms such as cobweb and honeycomb, but there is still a lot of room for improvement.

This paper designs a new multiple folded-beam disk resonator, which inherits the advantages of the radially pleated design, and also introduces lumped mass layers with better adaptability to the frame structure. The measured *Q* after processing reaches 710 k, proving to be a record in flexure silicon disk resonators, showing great potential for MEMS gyroscopes application.

## 2. Device Design and Modeling

Based on the widely used Zener’s standard model of TED [17]
(1)$$Q_{\mathrm{TED}}=\frac{C_{v}}{E\alpha^{2}T_{0}}(\frac{2\pi f_{0}}{f_{Z}}+\frac{f_{Z}}{2\pi f_{0}})$$
(2)$$f_{Z}=\frac{\pi^{2}\chi }{b^{2}}$$
where *C_v_* is the heat capacity at constant volume, *E* is the Young’s modulus, *α* is the linear coefficient of thermal expansion, *T*_0_ is the equilibrium temperature, *f*_0_ is the resonant frequency, and *f_Z_* is the thermal relaxation rate related to the thermal diffusivity *χ* and the strained beam width *b*. For most silicon-based MEMS resonators, smaller beam width can increase the thermal relaxation rate and reduce the resonant frequency, thus separate *f*_0_ from *f_Z_* and realize high *Q*_TED_. Considering the limitations of processing technology and the requirement of resonator stiffness, *b* cannot be infinitely reduced, and there will be a lower limit. Then we can introduce lumped masses to further decrease *f*_0_ without affecting *f_Z_* [12]. However, the maximization of this stiffness-mass decoupling effect is usually limited to the non-ideal compatibility between the frame and masses. Directly hanging masses on the outer rings also has a certain impact on the equivalent capacitance area and mechanical sensitivity of the resonator. Here, we propose another design idea. We carry out the integrated design for the multiple folded beams and the lumped mass layers from the beginning to gain great structure coherence. Then we further decrease the beam width and increase the mass width under the premise of ensuring rigidity. This trick can make full use of the stiffness-mass decoupling effect, thus provide lower *f*_0_ and higher *fz* to realize greater *Q*_TED_.

The multiple folded-beam disk resonator we designed is shown in Figure 1a. The resonator consists of a central anchor, multiple folded beams extending radially, and outer mass layers hanging between the beams. The inner folded beams with small width (*b*) can effectively separate resonance frequency and thermal relaxation frequency. With integral design, the outer mass layers help to further increase the effective mass and reduce the resonant frequency. Besides, the structure vibrates approximately parallel at resonance, which is favorable for increasing the equivalent capacitance area. The diameter of the resonator (*D*) is 6080 μm, the structure height is 100 μm, the anchor diameter (*d*) is 2420 μm. The thicknesses of outer mass layers (*t_m_*) are 80 μm. The gap between the electrodes and the resonant structure is 10 μm. The connecting-beam width (*b*), which shows an important impact on *Q*_TED_, is designed from 5.5 μm to 7.5 μm. Considering the displacement coupling of in-plane and out-of-plane deformations in (111) silicon, we selected (100) silicon for fabricating and chose the *n* = 3 wine-glass working mode, accordingly. Note that the Young’s modulus and Poisson’s ratio of (100) silicon is anisotropic, but holding a 90° rotation symmetry. If we rotate the mode shapes of the cos 3θ mode clockwise by an angle of 90, the orientations of the mode shapes relative to the Young’s modulus curve are exactly the same as those for the sin 3θ mode [18,19,20]. This ensures that the *n* = 3 modes satisfy frequency matching theoretically, but with distinct mode-shape asymmetry. The *n* = 3 mode shapes of the multiple folded-beam disk resonator are shown as Figure 1b, simulated with a density of 2330 kg/m^3^, thermal expansion coefficient of 2.6 ppm/K and the orthotropic stiffness matrix for (100) silicon with three axes at [100], [010], and [001] crystal orientations at 273 K. The amplitude of radial motion at the peaks along <110> orientation is distinctly different from that of the two symmetrical peaks.

The dynamical parameters of the proposed multiple folded-beam disk resonator (with a beam width of 6 μm) are calculated and compared with those of a honeycomb disk resonator, which have gone through the comprehensive optimization of structural parameters [16]. The resonant frequency *f*_0_ and thermoelastic quality factor of the working modes are simulated using COMSOL 5.5 with anisotropic (100) silicon. The effective mass *m*_eff_ and the Coriolis coupling factor *κ* can be calculated based on the following models [17,18].
(3)$$m_{\mathrm{eff}}=\underset{V}{∭}\rho (\phi_{X1}^{2}+\phi_{Y1}^{2}+\phi_{Z1}^{2})dV,$$
(4)$$\kappa =\frac{\underset{V}{∭}\rho (\phi_{X1}\phi_{Y2}-\phi_{X2}\phi_{Y1})dV}{m_{\mathrm{eff}}},$$
where (*ϕ_Xj_, ϕ_Yj_, ϕ_Zj_*) are the shape functions of the *j*th (*j* = 1, 2) degenerate modes in geometric coordinates *X*-*Y*-*Z*, and *ρ* is the density of monocrystalline silicon. The angular gain factor can also be obtained using *A_g_* = *κ/n*, where *n* = 3 in this study.

The calculated parameters are summarized in Table 1. Their working frequencies and effect masses compare closely. However, the simulated *Q*_TED_ of the multiple folded-beam disk resonator is about 2.27 times larger than that of the Honeycomb resonator.

The fabrication process of the resonator is shown in Figure 2a. Firstly, etch a substrate silicon-on-insulator (SOI) wafer for 10 μm to form the anchors and generate the thermal oxide layer on it with thermal oxidation technology. Then the substrate and another structure SOI are bonded together through wafer fusion-bonding technology. The handle wafer is moved with chemical and mechanical polishing subsequently. Next, the aluminum wire bonding pads are patterned, and the resonator and the electrodes are formed via deep reactive ion etching (DRIE) technology. Lastly, the device is diced by laser stealth cutting, attached to a chip carrier, wire-bonded and then sealed in a metal shell with a high vacuum of 0.001 Pa. The microscope images of the resonator are shown in Figure 2b.

## 3. Resonator Characterization

The resonant frequencies and dissipations of the operational *n* = 3 normal modes of the processed resonator were tested using the setup shown as Figure 3a, among which the double-pole double-throw switch S2 was used to change the testing axis. The frequency responses of the packaged resonator are shown in Figure 3b. Take a resonator with 6 μm-beam-width for example, the initial resonant frequencies of *n* = 3 normal modes are extracted to be *ω*_1_ = 2π × 4116.55 Hz and *ω*_2_ = 2π × 4118.24 Hz, respectively, indicating the initial frequency split is only 1.69 Hz. This shows that the *n* = 3 modes of the (100) silicon resonator hold good frequency-matching characteristics.

Then ring-down tests were implemented to characterize the quality factor *Q* accurately and the decaying time constant *τ* of the resonator. The resonator was initially actuated at resonance with a constant amplitude using a phase-locked loop (PLL) and a proportion-integration-differentiation (PID) amplitude controller. Then the actuation was stopped by turning off the switch S1. The measured decaying time constants were *τ*_1_ = 54.5 s and *τ*_2_ = 55.4 s, corresponding to *Q*_1_ = 704 k and *Q*_2_ = 716 k. This proved to be a new record for flexure silicon disk resonators.

## 4. Results and Discussion

The simulated *Q*_TED_ and the experimental *Q* of the fabricated resonators with different beam widths are shown in Figure 4a. With the decrease in beam widths, the simulated *Q*_TED_ (blue dots) could obviously be elevated, revealing that the *Q*_TED_ of the 5.5μm-beam-width was about twice that of the 7.5μm-beam-width. However, the difference of the measured *Q* between them was small, violating the distribution of *Q*_TED_. Neglecting the air damping in the high-vacuum metallic shell, the experimental quality factor *Q* of the resonator could be estimated by 1/ *Q* = 1/ *Q*_TED_ + 1/ *Q*_other_, where *Q*_other_ was used for calibration which represented the sum of the other damping mechanisms. At room temperature, *Q*_other_ of resonators with different beam widths was extracted as the red dots in Figure 4a, which was almost constant and ranged from 905 k to 1048 k. 

Then we conducted a temperature experiment for the 6 μm resonator whose quality factor was *Q* = 716 k at room temperature. The relation between the quality factor and temperature is shown in Figure 4b. The yellow dots are the measured overall quality factor *Q*, which increased before the temperature dropped to around −5 °C while gradually decreasing with continuing cooling. The maximal *Q* is 722 k, close to that measured at normal temperature. Similarly, based on the simulated *Q*_TED_ (blue dots), we can calculate *Q*_other_ shown as the red dots in Figure 4b, which changed between 835 k and 1020 k and was almost equal to that in Figure 4a. These results indicated that the dominating damping mechanism is no longer TED.

Note that though the (100) silicon for manufacturing was displacement-decoupled in inner and outer plane, the in-plane anisotropy of the material caused distinct modal asymmetry. For *n* = 3 mode of the multiple folded-beam disk resonator shown as Figure 5a, the amplitude of radial motion at the distinct minimal peaks along <110> orientation was 0.84 relative to that at the two symmetrical peaks, based on the simulated mode shape. Compared to the circumferentially symmetrical vibration in isotropic silicon resonators depicted as Figure 5b, this asymmetry along the circumference may further cause the anchor vibration in <110> orientation and aggravate support loss [21]. However, unlike the intractable out-of-plane displacement in (111) silicon, the asymmetrical in-plane displacement in (100) silicon resonator may be suppressed, through electrostatic tuning [22,23] or structural compensation [24]. Verification of the above inference is being conducted by us.

## 5. Conclusions

In this paper, a multiple folded-beam disk resonator with high *Q*_TED_ (more than 2 million) was demonstrated. The *n* = 3 modes of the (100) silicon resonator also held good frequency-matching characteristics. The quality factor of the fabricated resonator reached 710 k, proving to be a record in flexure silicon disk resonators. Moreover, conducted temperature experiment indicated that TED is no longer the dominating damping mechanism. As several researches have mentioned, the distinct mode asymmetry of the resonator caused by the in-plane anisotropy of material may lead to anchor vibration and exacerbate the anchor loss to limit the total quality factor. Future study on verifying the inference and compensating the mode asymmetry of this resonator is under way.

## Figures and Tables

**Figure 1 micromachines-13-01468-f001:**
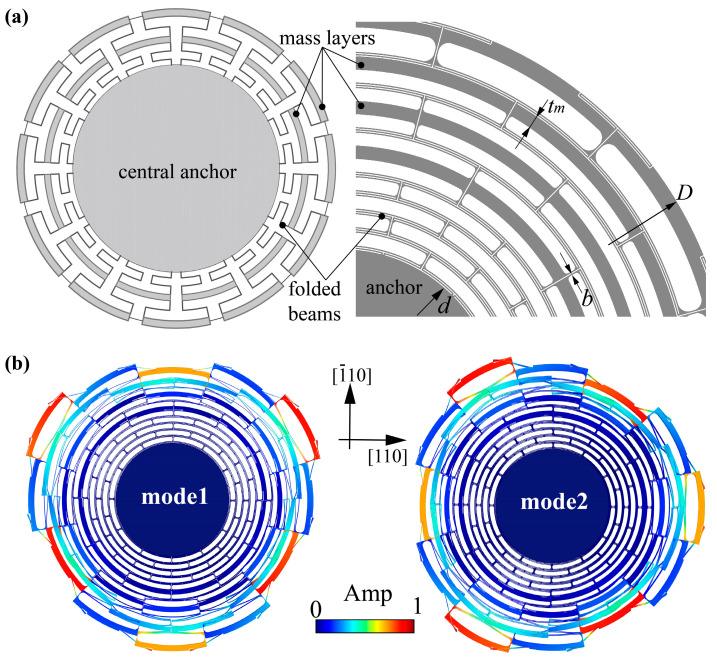
Design of the multiple folded-beam disk resonator. (**a**) Design concept and structural design of the multiple folded-beam disk resonator. (**b**) The displacement-normalized mode shapes obtained by simulation.

**Figure 2 micromachines-13-01468-f002:**
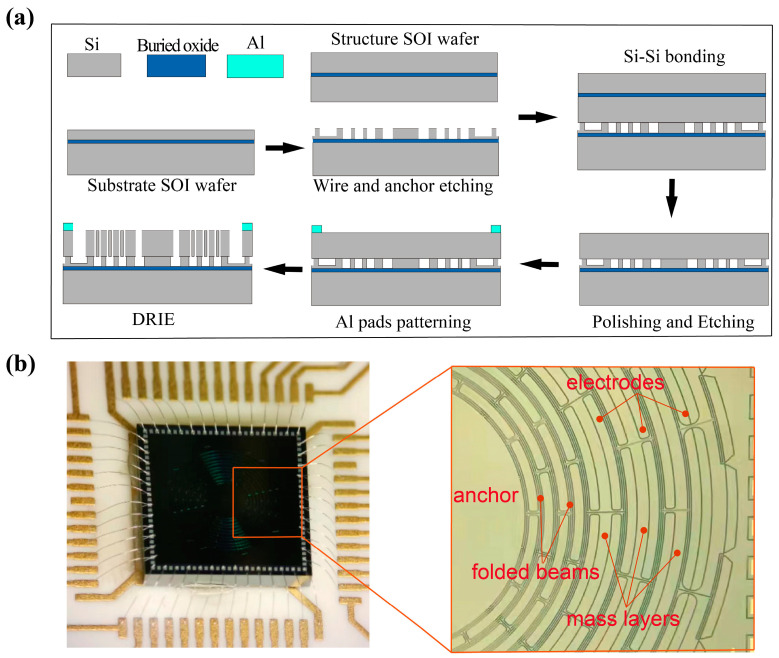
(**a**) Fabrication processes and (**b**) Microscope images of the resonator.

**Figure 3 micromachines-13-01468-f003:**
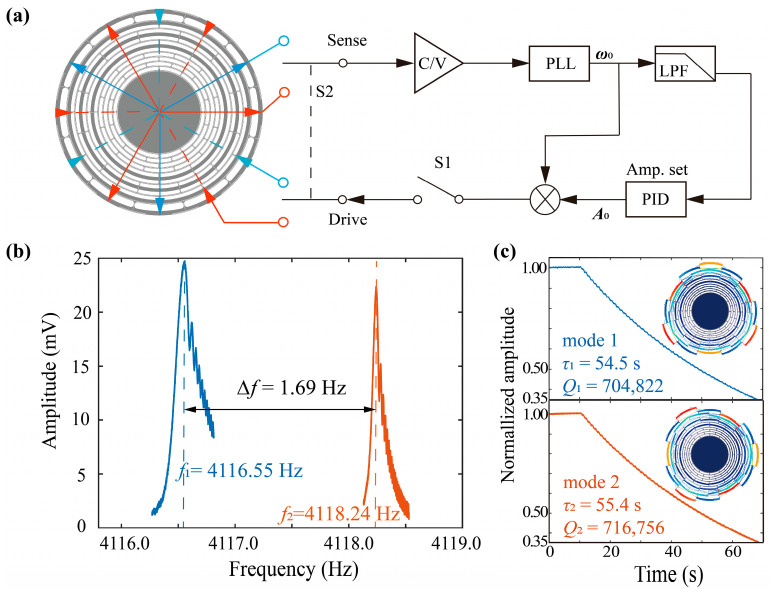
Test system and results. (**a**) Block diagram of the ring-down test circuit; (**b**) Frequency sweep curves and (**c**) Ring-down curves of the driving and sensing modes of the resonator with 6 μm-beam-width.

**Figure 4 micromachines-13-01468-f004:**
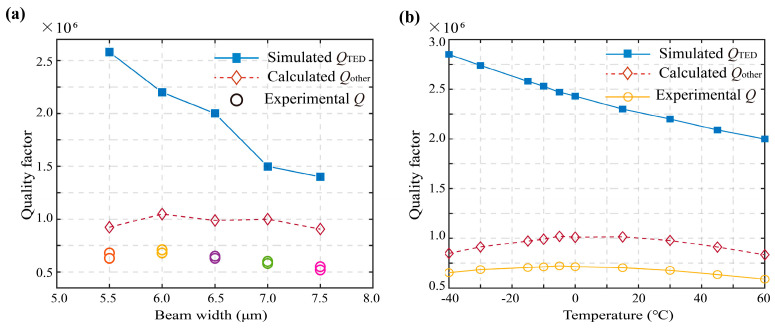
(**a**) Relation between quality factor and beam width. (**b**) Relation between quality factor and temperature.

**Figure 5 micromachines-13-01468-f005:**
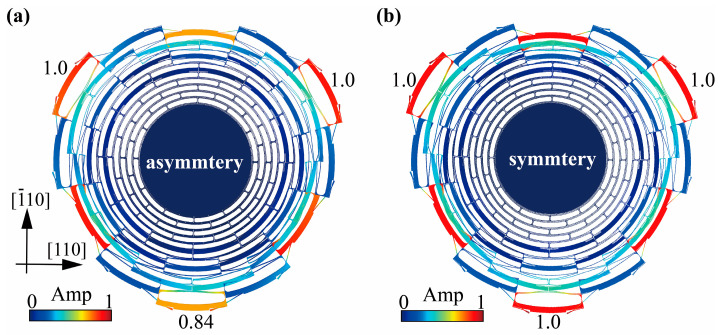
Characterization of the asymmetrical in-plane displacement in (**a**) (100) silicon resonator owing to the anisotropy of the material and (**b**) isotropic silicon resonator.

**Table 1 micromachines-13-01468-t001:** Comparison of the simulated parameters of different disk resonators.

Resonator Type	Honeycomb Resonator	Multiple Folded-Beam Disk Resonator
*f*_0_ (Hz)	4150	4231
*Q* _TED_	970 k	2.2 million
*m*_eff_ (mg)	1.75	1.61
*A* _g_	0.84	0.86

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
