# Peer review of "Design of a Multiple Folded-Beam Disk Resonator with High Quality Factor"

_micromachines, 2022, doi:10.3390/mi13091468_

Round 1

Reviewer 1 Report

This article presents progress in improving the mechanical quality factor for MEMS devices, which is an hot topic for micro and nano mechanical devices. The novelty comes from a specific design of the resonator which is optimized for reducing thermoelastic losses.

Experiment show qualitative agreement with COMSOL simulations, but as authors themselves aknowledge, simulations predict event higher quality factors. In order to understand the origin of this difference, I suggest Fig. 4 cto be completed with an 1/Q = f(beam width) plot, as different contributions to dissipation sum as parallel dampers. Such a plot will show an almost constant difference between simulation and experiment except for the first point (width = 5.5 microns).

I also have a question concerning Fig. 3. Why the width of the resonances presented on 3b does not correspond to the ring-down times presented on Fig. 3c? For example, if we take the width of the lower resonance on Fig 3b, it is higher than 0.1Hz, which corresponds to Q<41,165.5. And why do we see multiple resonances for both modes on Fig. 3b?

Last, I wold have a comment about the conclusion part. Including the Si anisotropy in elastic properties is clearly a path that is worth investigating for future COMSOL simulations and device designs; but for me this study does not prove that the Si anisotropy is the major source of discrepancy between COMSOL simulations and experiment, as stated by authors at lines 160-163 in the Conclusions part. I believe this sentence should either be changed or supported by more studies, like COMSOL simulations including the Si anisotropy.

Author Response

We appreciate Reviewers very much for their valuable suggestions and constructive comments on our manuscript. We have studied the comments very carefully and have tried our best to revise our manuscript according to the comments. Those changes will not influence the framework and the overall conclusion of this paper. Attached please find the revised manuscript and a detailed point-by-point response letter, which we would like to submit for your kind consideration.

Reviewer 2 Report

The Q factor of 710k for a silicon disk resonator is impressive and it is good news for DRGs. I have several questions to discuss with the authors.

1. Separating f0  from fis a consensus. Does the author have any special tricks to do that? 

2. In Figure 3(b), why there are so many sawteeth on the left of the response curves?

3. The experimental Q is different from the simulated QTED. The authors believe that some other factors dominate the Q.  However, the authors can verify the hypothesis by a temperature experiment since the QTED is related to temperature. By the way, if some other factors instead QTED dominate the Q, it seems futile to optimize the beam width of the resonator.

4. The Young's modulus in <100>and <110> is very different. How did the author match frequency split to 1.69 Hz? If employing the electrostatic tuning method, Please give the parameter, such as tuning voltage and tuning capacity.

5. Does the author consider the anisotropic of silicon when calculating the QTED since Young's modulus, thermal diffusivity, et.al would vary in different crystal orientation

6. The picture in Figure 1(a) and Figure 1(b) is not clear, and it makes me hard to understand the topological structure of the resonator. Please make it easy to understand.

Author Response

(The authors gave the same response as above.)

Round 2

Reviewer 2 Report

The author clears the doubts. I recommend publishing the manuscript.